# Willingness to Compromise Scale: Italian Validation and Assessment of the Relationship with Career Decision Self-Efficacy and Career Adaptability during School-to-Work Transition

**DOI:** 10.3390/ijerph20032662

**Published:** 2023-02-01

**Authors:** Anna Parola

**Affiliations:** Department of Humanities, University of Naples Federico II, 80133 Naples, Italy; anna.parola@unina.it

**Keywords:** willingness to compromise, validation study, second-order factorial analysis, career adaptability, career decision-making self-efficacy, school-to-work transition

## Abstract

Willingness to compromise is defined as the propensity to accept an alternative career-related option that was not the one initially desired. In the literature, there is a validated scale for measuring willingness to compromise but not an Italian validation. Thus, Study 1 aimed to test the psychometric proprieties of the Willingness to Compromise Scale in a sample of 282 Italian university students. Confirmatory factor analyses were performed showing a second-order factorial structure with two well-separated first-order factors, i.e., compromising and adapting. Study 2 focused on the predicting role of willingness to compromise on career decision self-efficacy and the mediating role of career adaptability in this relationship. The sample consisted of 237 Italian university students. A mediation analysis with a 5000-bootstrap resampling procedure was computed. The results showed that willingness to compromise predicts both career decision self-efficacy and career adaptability, while career adaptability mediates the relationship between willingness to compromise and career decision self-efficacy. These findings allowed the discussion of practical implications for career guidance intervention aimed to support school-to-work transitions.

## 1. Introduction

Making career choices is the most demanding developmental task for adolescents and young adults. Guided by life-span developmental psychology, the school-to-work transition is conceived as a make-or-break period, in which young people try to balance pressures for completing their educational qualifications and establishing themselves in the labor market [1], as well as other transitions that establish likewise crucial developmental outcomes such as moving away from the parental home and starting a family [2].

The features of the current labor market force a careful new reflection on career transitions [3]. Three main challenges affected and currently affect the educational system transitions and the school-to-work transitions. First, digitalization that, on one hand, has put some job sectors at risk and, on other hand, has required new skills and competences for “new” jobs. Second, the environmental challenges that have changed the “view” of jobs demanding specific professional abilities. Third, the economic recession that has reduced the possibility of the labor market, increasing unemployment. Young people have been most affected by the economic crisis: the latest data about youth unemployment [4] showed that in the EU about 12.7 million young people aged from 20 to 34 are neither in employment nor education and training (NEET).

Therefore, social and political phenomena must also be considered when trying to decipher the difficulties faced by young people in their transition to the world of work. In the last two years, pandemics and wars have characterized our scenario. The COVID-19 pandemic has affected daily life, the relational and school/work life. As reported by the International Labor Organization [5], among the different cohorts, young people represented the most vulnerable group to the economic consequences of COVID-19. Almost inevitably, the impacts of COVID-19 intersected with the precarious labor market. Indeed, social restrictions and lockdowns have led to the dismissal of precariously employed workers. Finally, war is also bringing changes to the global economy. The International Labor Organization [6] stated that we are going through “a set of multiple and overlapping crises, compounded by the Ukraine war and subsequent negative spill over effects, have materialized over 2022 which are deeply impacting the world of work”, and these conditions will also reduce the job demand. Moreover, the pandemic and war have exacerbated the already existing school-to-work transition difficulties [7]. This may be especially true in countries such as Italy, where contextual factors had a dramatic effect on the career transitions of adolescents and young adults. According to Eurostat [4], nine EU countries recorded rates of NEETs (young people aged 15–29 years not in employment or in education or training) above the EU average of 13.1% in 2021. Among them, the highest rates were recorded in Italy (23%).

Italian studies showed that university students feel uncertainty and instability in the labor market [8]. Moreover, non-adaptive transitions [3] affect developmental outcomes, such as the transition to adulthood [9,10], and health outcomes [11]. This situation has been exacerbated by the COVID-19 pandemic which has undermined the career aspirations of Italian university students [7,12,13,14].

In making decisions about their careers, young people often have to make compromises. These compromises may depend on internal barriers, such as the desired career requiring too much effort to obtain, or external barriers, such as contextual or environmental hindrances. Willingness to compromise is defined as the propensity to accept an alternative career-related option that was not the one initially desired [15]. This construct is different from career compromise, which, in turn, is related to accommodating a less-than-ideal career option [16]. Willingness to compromise is conceptualized as a stable individual difference that describes how a person might respond to various career-related decisions when they arise. According to the literature, this stable trait would allow us to capture the response (in terms of career choices) of individuals when faced with new and different situations [17]. 

Recognizing the need for a specific instrument to study career-related compromises, Wee [15] developed and validated a measure called the “Willingness to Compromise Scale”. A total of 171 individuals living in Singapore participated in the validation study. The final version of the instrument contains nine items. As Wee [15] argues, the items were generated to assess willingness to compromise both in terms of compromising (with a reverse code) and adapting. The items were rated on a 5-point Likert scale, from 1 (=strongly disagree) to 5 (=strongly agree). As an analytic strategy, the number of factors was determined through parallel analysis [18] and the minimum average partial criterion [19] revealed that the one-factor solution was appropriate. 

Good reliability and predictive values for psychological variables were confirmed in a later study. Specifically, the scale showed associations with other career constructs, such as dealing with uncertainty, openness to experience and career adaptability [15]. Creed and colleagues [20] have studied the role of willingness to compromise as defending or self-protecting behavior to preserve vocational identity homeostasis. The findings revealed that willingness to compromise partially mediated the relationship between vocational identity and career goal–performance discrepancy.

In the context of uncertainty described above, a measure for understanding willingness to compromise could also be useful in the Italian context wherein the school-to-work transition appears increasingly difficult for young adults. Since no standardized and validated version of the Willingness to Compromise Scale is available for the Italian population, the present study aimed to explore the factor structure of the Italian version of the Willingness to Compromise Scale (Study 1) and to assess its relationship with career decision self-efficacy and adaptability (Study 2).

Career adaptability and decision-making self-efficacy are crucial resources for dealing with future career obstacles and mastering career transitions such as school-to-work transitions in challenging environments [21,22].

According to the career construction theory (CCT; [23,24]), career adaptability is a fundamental prerequisite for managing career choices and successful school-to-work transition [25]. It is defined as a “psychosocial construct that denotes an individual’s resources for coping with current and anticipated tasks, transitions, traumas in their occupational roles” [21], p. 662. Four psychological skills, namely control, confidence, concern, and curiosity, have been identified as components of career adaptability [21]. Savickas [23] outlined that career adaptability involved readiness to cope with the unpredictable adjustments provoked by changes in the world of work. This concept is allied, but not overlapping, with the construct of willingness to compromise because the latter indicates the attitudes and beliefs an individual has when faced with a gap between his/her objective reality and planned goals. Guided by the CCT [23,24], adaptive readiness, adaptability resources, adapting responses, and adaptation results are empirically different from one another [26,27], and some studies have confirmed this distinction [28,29,30]. The willingness to compromise seems to refer more to adaptive readiness than adaptability resources which refers, instead, to the four career adaptabilities that help individuals cope with current or anticipated change and are self-regulation strengths or capacities [21,23].

Following the conceptual and empirical investigation of Hirschi and colleagues [26], the willingness to compromise as a context-general adaptivity trait (also called readiness) might predict career adaptability which, in turn, predicts career responses such as career decision-making self-efficacy.

To adapt to one’s contexts in making career decisions, the individual also needs career decision-making self-efficacy [31]. Career decision-making self-efficacy is defined as the individual’s belief that he/she is successful in completing decision-making tasks associated with his/her career [32]. In this regard, career decision self-efficacy may refer to an adaptive response, which, in turn, can affect adaptation results such as promoting career exploration [32,33], and/or success in an adaptive school-to-work transition. Previous studies provided important evidence regarding the effects of career decision-making self-efficacy, which functions as a significant mediator between several psychological variables and career adaptability, such as personality affected [34], social support [35], emotional intelligence [36], and self-esteem [36].

While the relationship between career adaptability and career decision-making self-efficacy has been researched in the literature (for a meta-analysis, see [37]), the relationship between willingness to compromise and career decision-making self-efficacy remains poorly investigated. Nevertheless, a good/high ability to compromise by effectively evaluating career options could favor feeling effective in making career choices. This study intends to relate willingness to compromise, career adaptability, and career decision-making self-efficacy as follows: assuming career adaptability as an adaptive resource, this study hypothesizes that career adaptability may be a powerful mediator in the relationship between willingness to compromise (adaptive readiness) and career decision-making self-efficacy (adaptive response).

Given this background, the present study has two aims. First, to assess the psychometric characteristics of the Willingness to Compromise Scale in the Italian context (Study 1). Second, assessing the willingness to compromise as a predictor of career decision-making self-efficacy, assuming a mediating role of career adaptability in this relationship (Study 2).

## 2. Study 1 

### 2.1. Materials and Methods

#### 2.1.1. Translation and Cross-Cultural Adaptation

Following the guidelines [38,39] and previous procedures of Italian validation studies [40,41,42], the Willingness to Compromise Scale was independently translated by two Italian experts in the career field and back-translated into English by an independent translator to guarantee cross-cultural equivalence.

The questionnaire was also administered to 20 Italian university students to assess whether the items were understandable by the target population. No changes have been made.

#### 2.1.2. Participants and Procedure

The subject-per-parameter ratio “n:q criterion” was used to plan a priori the minimum number of subjects needed. A ratio of 5 subjects per parameter was guaranteed [43,44,45,46].

The sample was composed of 282 Italian university students (102 males and 180 females) aged from 18 to 31 (M = 24.86; SD = 4.12). The snowball sampling method was used. To ensure coverage of the different regions of Italy, participants were recruited from the general population through advertisements on social media (e.g., Facebook, Twitter, etc.). Participants were distributed as follows: 28.8% north; 24% center; 40.7% south; 6.5% islands.

Inclusion criteria were: (A) being over 18 years, (B) being a university student, (C) being a native Italian speaker, and (D) providing informed consent.

#### 2.1.3. Measures

Data were collected using Google Forms. Participants voluntarily accessed the online platform and were not offered any incentive or compensation for participating in this study.

A biographic information form collected general demographic information (e.g., sex, age, civil status, and residence).

In addition, the Willingness to Compromise Scale were administered.

*Willingness to Compromise Scale.* This instrument consists of 9 items assessing the willingness to compromise. Participants responded to each item on a 5-point Likert scale from 1 (=strongly disagree) to 5 (=strongly agree). Higher scores indicate a greater level of willingness to compromise. Example of items are: “Reality constraints should not stand in the way of one’s career goals” (reverse coded), “Once I decide on a desired career outcome, no other career outcome would be acceptable” (reverse coded). The internal reliability of the original scale was 0.77. In this study, the Italian translation version was used.

#### 2.1.4. Statistical Analysis

To test the factorial structure of the Willingness to Compromise Scale, a confirmatory factor analysis (CFA) was performed. According to the original validation study, a single-factor model was specified (Model 1). Before the computation, items 1, 2, 5, 6, 7, 9 were reversed as indicated in the original study. No missing data were found in the dataset.

The ML estimator was used to perform the confirmatory factor analysis (CFA). 

To evaluate the adequacy of models for the data, the chi-square statistic, the CFI, the RMSEA with associated 90% confidence intervals, and SRMR were used. The following cut-off criteria were chosen to evaluate the goodness of fit: (a) statistical non-significance of the χ^2^, (b) an RMSEA lower than 0.08, (c) a CFI higher than 0.90, and (d) an SRMR lower than 0.08 [43,44,47,48].

The internal consistencies of factors were evaluated by computing Cronbach’s alpha (α).

## 3. Results

A single factor model (Model 1) showed a non-adequate fit: χ^2^ (27) = 156.297, RMSEA (0.130; 90% CI 0.111–0.150), CFI (0.789) and SRMR (0.095).

Thus, considering the theoretical framework [15], as well as the semantic content of the items, a second-order factorial structure (Model 2) was specified and tested as follows: each item was loaded onto its specific first-order factors reflecting the main two domains of willingness to compromise—namely, (A) “compromising”, (B) “adapting”—and an overarching general factor called “willingness to compromise” (Figure 1). 

Model 2 showed an adequate fit to the data. Even if the chi-square statistic was still statistically significant (χ^2^ (25)  =  62.770, *p* < 0.001, the RMSEA (0.073; 90% CI 0.052–0.095), the CFI (0.927), and the SRMR (0.074) revealed a good model fit. As displayed in Table 1, all items’ loadings were statistically significant and ranged from 0.445 (item 5) to 0.795 (item 1).

Considering the internal consistency, Cronbach’s alpha revealed that the willingness to compromise showed good internal consistency for each domain, 0.789 for compromising and 0.737 for sadapting, and a general total score of 0.765.

## 4. Study 2

### 4.1. Materials and Methods

#### 4.1.1. Participants and Procedure

The sample was composed of 237 Italian university students (45 males and 192 females) aged from 18 to 30 (M = 22.35; SD = 3.05). In line with the first study, the snowball sampling method was used. To ensure coverage of the different regions of Italy, participants were recruited from the general population through advertisements on social media (e.g., Facebook, Twitter, etc.). Participants were distributed as follows: 22.2% north; 28.2% center; 45.8% south; 3.8% islands.

Inclusion criteria were: (A) being over 18 years, (B) being a university student, (C) being a native Italian speaker, and (D) providing informed consent.

#### 4.1.2. Measures

As in Study 1, data were collected using Google Forms. Participants voluntarily accessed the online platform and were not offered any incentive or compensation for participating in this study.

The socio-demographic information form used in Study 1 and the Willingness to Compromise Scale were administered. For this study, the total score was used. Higher scores represent a greater level of willingness to compromise. The Cronbach’s alpha of the total score was 0.787.

In addition, the following self-report measures were administered.

*Career decision self-efficacy*. Career decision self-efficacy was measured with the Career Decision Self-Efficacy Scale Short Version [49,50]. The measure consists of 25 items rated on a 5-point Likert scale ranging from 1 (=not at all confident) to 5 (=totally confident). 

This instrument includes five dimensions: self-appraisal (5 items, e.g., “Decide what you value most in an occupation”), occupational information (5 items, e.g., “Talk with a person already employed in the field you are interested in”), goal selection (5 items, “Choose a career that will fit your preferred lifestyle”), planning (5 items, e.g., “Make a plan of your goals for the next 5 years”), problem solving (5 items, e.g., “Identify from reasonable major (field of study) or career alternatives if you are unable to get your first choice”). For this study, the total score was used. Higher scores represent a greater level of career decision self-efficacy. The Cronbach’s alpha of the total score was 0.956.

*Career adaptability.* Career adaptability was measured with the Career Adapt-Abilities Scale [21,51]. The measure consists of 24 items rated on a 5-point Likert scale ranging from 1 (=not strong) to 5 (=strongest). 

This instrument includes four dimensions: concern (6 items, e.g., “Planning how to achieve my goals), control (6 items, e.g., “Making decisions by myself”), curiosity (6 items, e.g., “Looking for opportunities to grow as a person”), and confidence (6 items, e.g., “Solving problems”). For this study, the total score was used. Higher scores represent a greater level of career adaptability. The Cronbach’s alpha of the total score was 0.959.

#### 4.1.3. Statistical Analysis

Pearson’s correlations between willingness to compromise, career decision self-efficacy, and career adaptability were computed. The mediation model was tested using a two-step approach [52,53]. 

Step 1: a predictor-only model was specified: the “willingness to compromise” (X) was regressed on “career decision self-efficacy” (Y). 

Step 2: the full mediation model was specified: the “willingness to compromise” (X) was regressed on “career decision self-efficacy” (Y) through “career adaptability” (see Figure 2).

A mediation analysis was computed with a 5000-bootstrap resampling procedure. The mediation analysis tested whether the indirect effect of career adaptability mediated the effect of willingness to compromise and career decision self-efficacy with the bootstrapping confidence interval. Considering the score distribution of the measured variables, the maximum likelihood (ML) estimator was used. All the reported regression coefficients were unstandardized (B). 

### 4.2. Results

Means, standard deviations, and correlation analysis are displayed in Table 2. A strong relationship emerges among all variables. As noted in Table 2, the analyses showed that career decision self-efficacy correlates positively with willingness to compromise (*r* = 0.558), and career adaptability (*r* = 0.767). In addition, a positive association emerges between willingness to compromise and career adaptability (*r* = 0.645). 

The mediation path analyzed whether the willingness to compromise predicted career decision self-efficacy and whether career adaptability mediated the effect of willingness to compromise on career decision self-efficacy. 

The direct effect (Figure 3, Table 3) showed a significant direct effect of willingness to compromise on career decision self-efficacy (B = 0.103, SE = 0.513, *p* = 0.046) and on career adaptability (B = 0.554, SE = 0.043, *p* < 0.001). 

Moreover, a significant direct effect of career adaptability on career decision self-efficacy (B = 0.763 SE = 0.059, *p* < 0.001) was found.

Bootstrapping analysis indicates that the indirect effect was significant (B = 0.423, SE = 0.062, *p* < 0.001, CI [0.599, 0.841]). 

The total indirect effect (willingness to compromise → career adaptability → career decision self-efficacy) was statistically significant (B = 0.526, SE = 0.510, *p* < 0.001, CI [0.426, 0.627]). The total explained variance (R^2^) was equal to 0.312.

## 5. Discussion

The present work aimed to test the psychometric characteristics of the Willingness to Compromise Scale in the Italian context. In Study 1, the factor structure and reliability of the Willingness to Compromise Scale were tested in a young adult population. To our knowledge, no previous study assessed the structural validity of the Willingness to Compromise Scale in the Italian context. Thus, the study aimed to cover this lack. In Study 2, the relationships between willingness to compromise, career decision self-efficacy, and career adaptability were evaluated.

In Study 1, the structure of the Italian versions of the Willingness to Compromise Scale was tested through CFA following the original one-factor structure [15]. However, the fit indices were not satisfactory. A different structure was specified as each item was loaded onto its specific first-order factors reflecting the main two domains of willingness to compromise—namely, (A) “compromising”, (B) “adapting”—and an overarching general factor called “willingness to compromise”. This model followed the theoretical background proposed initially by Wee [15], which guided the original development of the items. Indeed, as Wee claimed, “items were generated to assess willingness to compromise in terms of both compromising (e.g., I would pursue my career goals even if there were only a small chance that I could achieve it [reverse-coded]) and adapting (e.g., “I would consider a different job from my intended job if it were not my desired career outcome”)” [15], (p.493). This second-order model revealed a good fit. Therefore, the Italian version of this instrument is composed of two dimensions, compromising (six items) and adapting (three items), and an overarching general factor.

Although with a different factor structure, the questionnaire is in line with the theoretical assumptions of the construct. Indeed, the solid and theoretically driven methodology was maintained. It proved to be a reliable and psychometrically sound assessment tool to measure the willingness to compromise scale in young adults—specifically focused on the two main domains of compromising and adapting. 

The Italian version of the Willing to Compromise Scale, therefore, proved to be valid for young adults in the Italian context. Moreover, the Willingness to Compromise Scale showed a second-order, i.e., hierarchical, factorial structure with two well-separated first-order factors—clearly reflecting the two main domains of willingness to compromise—providing good fit indices. All the items had good factorial loadings on the hypothesized factors. Furthermore, the Willingness to Compromise Scale allowed observing how willingness can have some aspects in common with the two components.

In Study 2, the relationships between willingness to compromise, career decision self-efficacy, and career adaptability were assessed through correlation analysis. The findings showed a strong relationship between career decision self-efficacy and willingness to compromise, revealing how an individual’s greater ability to compromise is linked with a greater perceived feeling of self-efficacy in making decisions affecting his/her future career choices. Moreover, a positive association emerges between willingness to compromise and career adaptability. This result is in line with Wee [15], suggesting that both career adaptability and willingness to compromise deal with cognitive and behavioral flexibility in the face of a changing environment. Finally, the association between career decision self-efficacy and career adaptability emerged. This result is in line with several previous studies [37]. 

A mediation path analysis was conducted to capture the effect of willingness to compromise on career decision self-efficacy and the role of career adaptability in this relationship.

It was hypothesized that career adaptability, seen as a resource, could mediate the relationship between willingness to compromise and career decision self-efficacy. The results showed that willingness to compromise predicts both career decision self-efficacy and career adaptability. This evidence suggested that when the individual has the ability to compare his/her career desires with actual career options, this increases perceived decision-making self-efficacy. Furthermore, the effect of the willingness to compromise on career decision-making self-efficacy increases as career adaptability increases. This mediation mechanism explains the process of adaptive readiness → adaptive resources → adaptive response.

Therefore, the results of Study 1 reveal the Willingness to Compromise Scale is also usable in the Italian context, while the results of Study 2 provide support for the willingness to compromise as a broad attitudinal construct and the possible career outcomes of willingness to compromise, such as career decision self-efficacy.

This study is not without limitations. Firstly, Study 1 did not investigate the invariance between genders. The number of young adults within the two different groups did not allow for the measurement invariance test [43,45,54]. Future investigation will explore the validity of the Willingness to Compromise Scale across male and female groups. For the same reason, Study 2 did not investigate the gender differences through a multi-group analysis comparing the model across males and females [45,47]. Secondly, the observational research design did not allow for defining a causal relationship among variables. Future longitudinal studies will be needed to assess the mechanism of career choices. Finally, the survey consisted of self-report measures that may have been influenced by well-known biases, such as social desirability.

Despite this limitation, these studies confirm the possibility to use the Willingness to Compromise Scale with young adults in the Italian context and provide the first evidence of the Willingness to Compromise Scale as a promising instrument for assessing the willingness to compromise. Studying willingness to compromise can help researchers to understand the mechanisms of career decisions and predict a range of behavioral responses. Moreover, assessing willingness to compromise can help young adults in their school-to-work transition and facilitate the related career decision-making process. This instrument may help career practitioners to assess how the individual interprets and responds to various career-related decisions and orient their career interventions.

Future studies should be directed at studying other possible outcomes of willingness to compromise as well as predictors. For example, it would be interesting to study the role of external influences, such as parents. Moreover, personality traits might also play a role in predicting willingness to compromise.

Finally, career guidance implications should also be considered. Findings suggest a potential line of intervention in order to offer psychological help for individuals facing the school-to-work transition and dealing with the adverse challenges of the current world of work. 

## Figures and Tables

**Figure 1 ijerph-20-02662-f001:**
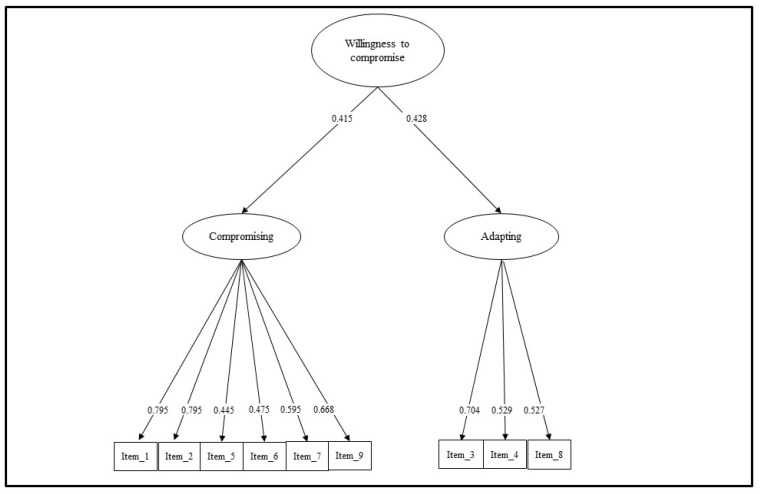
Second-order factor model.

**Figure 2 ijerph-20-02662-f002:**
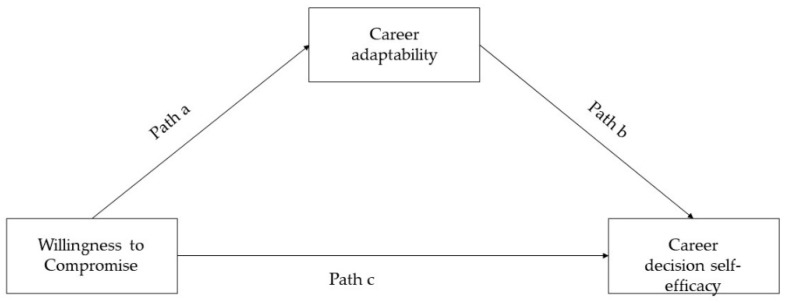
Statistical Diagram.

**Figure 3 ijerph-20-02662-f003:**
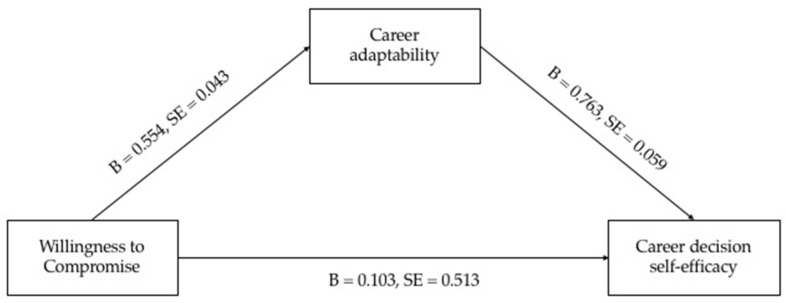
Mediation model.

**Table 1 ijerph-20-02662-t001:** Item descriptive statistics and confirmatory factor analysis.

	Descriptive Statistics				CFA	
	Mean	SD	SK	K	λ	*R* ^2^
Item 1	4.100	0.852	−0.616	−0.400	0.795	0.367
Item 2	4.210	0.799	−0.782	0.268	0.795	0.369
Item 5	4.050	0.891	−0.553	−0.462	0.445	0.504
Item 6	2.790	1.059	0.344	−0.355	0.475	0.720
Item 7	3.670	0.948	−0.222	−0.633	0.595	0.802
Item 9	3.960	0.918	−0.563	−0.386	0.668	0.775
Item 3	3.490	0.925	−0.254	−0.133	0.704	0.646
Item 4	2.930	1.041	0.009	−0.589	0.529	0.643
Item 9	3.960	0.918	−0.563	−0.386	0.597	0.554
Compromising	3.797	0.638	−0.166	−0.378	0.415	0.828
Adapting	3.348	0.727	0.065	−0.187	0.428	0.817
Willingness to compromise	3.572	0.511	0.537	0.620		

Note. CFA = confirmatory factor analysis. In the CFA columns, absolute values of standardized factor loading (|λ|) are reported. λ *=* factor loading onto the specific factor (i.e., compromising or adapting); for compromising and adapting, λ *refers to* factor loading of the first-order factors onto the general factor (i.e., “willingness to compromise”).

**Table 2 ijerph-20-02662-t002:** Means, standard deviations, αs, and correlations of willingness to compromise, career decision self-efficacy, and career adaptability.

		α	M	SD	1	2	3
1	CDSES	0.956	4.020	0.598	-		
2	WCS	0.787	3.759	0.634	0.558	-	
3	CAAS	0.959	4.288	0.545	0.767	0.645	-

Note. α = Cronbach’s alpha; M = mean; SD = standard deviation; *p* < 0.001.

**Table 3 ijerph-20-02662-t003:** Mediation model—direct, indirect, and total effects.

	B	Se	95%CI [L-U]
Path a	0.554	0.043	[0.470; 0.639]
Path b	0.763	0.059	[0.645; 0.881]
Path c	0.103	0.513	[0.002; 0.204]
Indirect effect	0.423	0.062	[0.599; 0.841]
Total indirect effect	0.526	0.510	[0.426; 0.627]

Note. B = unstandardized regression coefficient, se = standard error; 95%CI = confidence interval at 95%. Total indirect effect = willingness to compromise → career adaptability → career decision self-efficacy.

## Data Availability

The data presented in this study are available on request from the corresponding author. The data are not publicly available due to privacy issues.

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
