# Peer review of "Willingness to Compromise Scale: Italian Validation and Assessment of the Relationship with Career Decision Self-Efficacy and Career Adaptability during School-to-Work Transition"

_ijerph, 2023, doi:10.3390/ijerph20032662_

Round 1

Reviewer 1 Report

Thanks for the opportunity to review the paper Willingness to Compromise Scale: Italian validation and assessment of the relationship with career decision self-efficacy and career adaptability during school-to-work transition. I enjoyed this piece and highly appreciated how the author addressed the gap in knowledge in this research literature. I recommend this to be publish after some minor revision.

Overall, the manuscript is well-written, and the methodology sounds robust. I have a few suggestions, as I documented below, to further strengthen argumentations and to streamline the text.

Throughout the manuscript, grammar checks and language revision are required. I will not list all, but a few examples are: line 116, witch instead of which; line 259, were displayed in the Table 2 instead are displayed in Table 2.

Method

line 155, seeing that you declared that you had ensured coverage of the different regions of Italy, please, provide percentage of population by area (i.e., North, Centre, South, Isles);

line 160 + line 218, please, provide more information on procedure (i.e., how did you collect data? online? through which platform? in person?);

line 166, I suggest to provide at least one example of the scale item

Author Response

Dear Reviewer 1,

thank you for spending your time to review my paper. I am glad for your positive comments.

I would like to point out that, in accordance with what you suggested, I have proofread the paper and fixed typos and unclear sentences, including, of course, the ones you pointed out to me. Thank you!

Below is the point-by-point response:

line 155, seeing that you declared that you had ensured coverage of the different regions of Italy, please, provide percentage of population by area (i.e., North, Centre, South, Isles);

Percentages of population by area were included for both studies. Thank you!

- line 160 + line 218, please, provide more information on procedure (i.e., how did you collect data? online? through which platform? in person?);

This information for both studies was included. Thank you

- line 166, I suggest to provide at least one example of the scale item

Examples of items have been included. Thank you!

Author Response

Dear Reviewer 1,

thank you for spending your time to review my paper. I am glad for your positive comments. Below is the point-by-point response:

  1. Introduction
  • Page 2, lines 85-86. Since the author refers to the Italian employment youth outlooks here, I'd suggest they add some references to the Italian situation in the general problem description made up in the paper’s opening. 

Thank you for this comment that allowed me to describe the Italian situation and present the literature on this topic that reports evidence in the Italian population. I have included this aspect at the beginning of the paper, as suggested.

  • Pages 2 and 3, lines 91-133. I'd suggest the author edit this part a little bit to clarify their intentions here. If I got it well willing to compromise is closer to readiness (adaptivity) than adaptability. Related to this, I guess they connect WTC to adaptability based on previous research (e.g. Hirschi et al., 2015) in which readiness/adaptivity predicts adaptability which, in turn, predicts responses (e.g. CDSE). To make a reader more comfortable with these relationships, I'd suggest that the author clearly and explicitly state that they not only base on the research differentiating the concept but also build on the direction of the relationships provided by that piece of research. 

Thank you for this comment which allowed me to better clarify the framework and directions assumed in the study.

  • 1 Materials and Methods
  • Page 7, line 257. shouldn't an unstandardised coefficient be labelled with "B" or "b" instead of “β”?

Yes. Thank you for pointing it out! I edited in the body text, Table 3 and figure 3